# The Effects of Antibiotic Combination Treatments on *Pseudomonas aeruginosa* Tolerance Evolution and Coexistence with *Stenotrophomonas maltophilia*

Jack P. Law,[a] A. Jamie Wood,[a,b] Ville-Petri Friman[a]

aDepartment of Biology, University of York, York, United Kingdom
bDepartment of Mathematics, University of York, York, United Kingdom

**ABSTRACT** The *Pseudomonas aeruginosa* bacterium is a common pathogen of cystic fibrosis (CF) patients due to its ability to evolve resistance to antibiotics during treatments. While *P. aeruginosa* resistance evolution is well-characterized in monocultures, it is less well-understood in polymicrobial CF infections. Here, we investigated how exposure to ciprofloxacin, colistin, or tobramycin antibiotics, administered at sub-minimum inhibitory concentration (MIC) doses, both alone and in combination, shaped the tolerance evolution of *P. aeruginosa* (PAO1 lab and clinical CF LESB58 strains) in the absence and presence of a commonly co-occurring species, *Stenotrophomonas maltophilia*. The increases in antibiotic tolerances were primarily driven by the presence of that antibiotic in the treatment. We observed a reciprocal cross-tolerance between ciprofloxacin and tobramycin, and, when combined, the selected antibiotics increased the MICs for all of the antibiotics. Though the presence of *S. maltophilia* did not affect the tolerance or the MIC evolution, it drove *P. aeruginosa* into extinction more frequently in the presence of tobramycin due to its relatively greater innate tobramycin tolerance. In contrast, *P. aeruginosa* dominated and drove *S. maltophilia* extinct in most other treatments. Together, our findings suggest that besides driving high-level antibiotic tolerance evolution, sub-MIC antibiotic exposure can alter competitive bacterial interactions, leading to target pathogen extinctions in multispecies communities.

**IMPORTANCE** Cystic fibrosis (CF) is a genetic condition that results in thick mucus secretions in the lungs that are susceptible to chronic bacterial infections. The bacterial pathogen *Pseudomonas aeruginosa* is often associated with morbidity in CF and is difficult to treat due to its high resistance to antibiotics. The resistance evolution of *Pseudomonas aeruginosa* is poorly understood in polymicrobial infections that are typical of CF. To study this, we exposed *P. aeruginosa* to sublethal concentrations of ciprofloxacin, colistin, or tobramycin antibiotics in the absence and presence of a commonly co-occurring CF species, *Stenotrophomonas maltophilia*. We found that low-level antibiotic concentrations selected for high-level antibiotic resistance. While *P. aeruginosa* dominated in most antibiotic treatments, *S. maltophilia* drove it into extinction in the presence of tobramycin due to an innately higher tobramycin resistance. Our findings suggest that, besides driving high-level antibiotic tolerance evolution, sublethal antibiotic exposure can magnify competition in bacterial communities, which can lead to target pathogen extinctions in multispecies communities.

**KEYWORDS** experimental evolution, interspecies interactions, cystic fibrosis

Cystic fibrosis (CF) is a genetic condition that is characterized by impaired chloride ion channel function, which results in thick mucus secretions in the lungs that are susceptible to chronic bacterial infection (1). Of the bacterial species that infect adult CF patients, *Pseudomonas aeruginosa* is the most prevalent pathogen associated with

Address correspondence to Ville-Petri Friman, ville.friman@york.ac.uk.

The authors declare no conflict of interest.

morbidity (2, 3), and it is difficult to treat due to its intrinsic resistance to many antibiotics and its ability to readily evolve resistance to new antibiotics (4, 5). Though they are usually dominated by *P. aeruginosa*, CF infections are often polymicrobial, and many different bacterial species co-occur with *P. aeruginosa* in CF lungs (6–10).

Over the courses of their lives, patients with CF will be treated with a number of different antibiotics, including those administered during treatment to eradicate *P. aeruginosa* or to help resolve pulmonary exacerbations (11, 12). Antibiotics are administered at a high concentration, multiple times per day, to maintain a therapeutic dose at a concentration that is greater than the minimum inhibitory concentration (MIC) required to inhibit bacterial growth (12–14). Antibiotic combinations are used to target multiple species simultaneously or to increase the efficacy against a single species (15, 16). However, the thick mucus secretions and complex branching structure of the lungs themselves will likely result in bacterial populations experiencing a gradient of antibiotic concentrations (17, 18). Thus, despite the best efforts of the treatment regimens, pockets of bacteria within the lungs are likely to experience antibiotic concentrations below those required to inhibit those bacteria, and such subinhibitory concentrations have been shown to promote the evolution of antibiotic resistance (19, 20).

Selection for antibiotic resistance differs between antibiotics administered at or above the MIC and antibiotics administered below the MIC (14). At concentrations greater than MIC, the driver of selection is whether the bacteria can survive the antibiotic challenge; thus, any mutations in the bacterial population that increase the MIC, regardless of their impact on other competitive growth traits, would be selected (19, 20). Conversely, below the MIC, the selective pressure differs such that any mutation that confers an increase in growth in the presence of the low antibiotic concentration, and thus confers a competitive advantage relative to the other members of the population, would be selected, regardless of whether this mutation would increase the MIC (19, 20). This relatively weaker selection pressure increases the number of viable mutations, which in turn increases the likelihood that one such mutation could increase the MIC through mechanisms not traditionally considered to be involved in resistance (20). The lack of antibiotic-mediated killing also results in a longer selective window, during which more mutations can accumulate and either ameliorate the costs of higher-level resistance (21) or together confer high-level resistance via epistatic interactions (22). While the effects of lethal concentrations (greater than the MIC) of antibiotic combinations on individual bacterial species have been explored previously (23, 24), they are less well-understood at sub-MIC levels in multispecies communities.

Competition with other bacterial species could change the trajectory of antibiotic resistance evolution in a focal pathogen species in various ways (25). First, the presence of competitors more tolerant of an antibiotic treatment than the susceptible pathogen species could increase the strength of the competition between the two and could lead to a decrease in the relative pathogen abundance, potentially triggering extinctions (26). Competitor-mediated reduction in the population density of the focal pathogen could further slow resistance evolution by reducing the mutation supply rate and the emergence of *de novo* resistance (27). When antibiotic resistance evolves, it is often associated with metabolic costs, such as the activation of efflux pumps or the modification of the antibiotic target. These costs could reduce pathogen competitiveness in the presence of nonresistant mutants or species that are unaffected by a given antibiotic when antibiotic concentrations are low (28–30). While it has been suggested that bacterial interactions are predominantly competitive (31), it is also possible that other interacting bacteria could facilitate the antibiotic resistance of the focal species via the horizontal gene transfer of resistance genes (32). Alternatively, other species could provide protection from antibiotics via secretions that break down antibiotics (21, 33, 34) or create protective microenvironments via the production of biofilms (30, 35). Despite bacterial interspecies interactions being a ubiquitous selective force in nature, there are relatively few studies that directly test their effects on the evolution of antibiotic resistance.

Here, we focused on studying how the evolution of the antibiotic tolerance of *P. aeruginosa* is affected by the presence of *Stenotrophomonas maltophilia*, another CF-

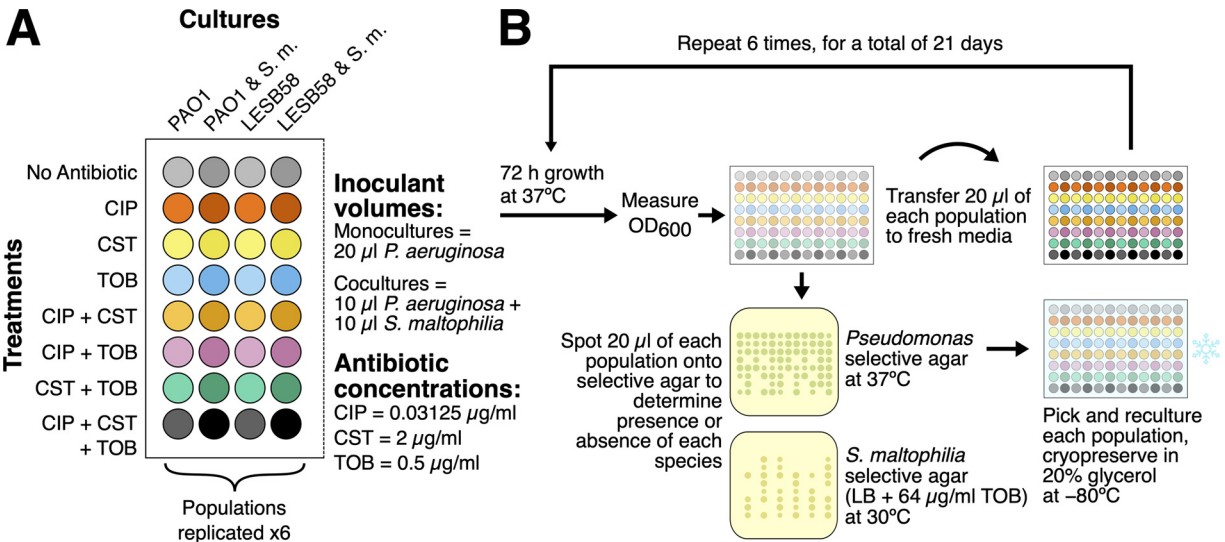

**FIG 1** Methods schematic. (A) The combinations of antibiotic treatments and bacterial cultures that were used during the selection experiment. (B) The procedure followed during the selection experiment.

associated species that is increasing in prevalence among CF patients (36–41) and that commonly co-occurs with *P. aeruginosa* (42). In order to investigate this, we performed a short-term *in vitro* serial transfer experiment in which we grew both the lung-naive laboratory *P. aeruginosa* strain PAO1 and the lung-adapted Liverpool epidemic strain B58 (LESB58) (42) either alone in monoculture or in the presence of *S. maltophilia* (resulting in four different cultures: PAO1, LESB58, PAO1 & *S. maltophilia*, and LESB58 & *S. maltophilia*) (Fig. 1). Two strains of *P. aeruginosa* were chosen to compare the potential effects of previous exposures to antibiotic treatments and other infecting bacteria on the evolution of antibiotic tolerance. Each of these cultures was treated with one of the eight combinations (see Materials and Methods) of the antipseudomonal antibiotics ciprofloxacin, colistin, and tobramycin. These antibiotics were selected because of their use either in *P. aeruginosa* eradication therapy or in the treatment of pulmonary exacerbations (11, 12, 43, 44) as well as for their differing modes of action (45–47). Each of the antibiotics was applied at a sub-MIC that had small but contrasting effects on the growth of all three bacterial strains (Fig. S1).

During the serial transfer experiment, which took place over 21 days, we tracked the presence of *P. aeruginosa* and *S. maltophilia* for any extinctions and monitored the changes in the total population densities across the 192 selection lines. Following the experiment, we measured the ability of the evolved focal *P. aeruginosa* isolates to grow in the treatment concentrations of the individual antibiotics, relative to ancestral stock strains. Moreover, the MICs of each antibiotic were determined for all of the evolved *P. aeruginosa* isolates. We hypothesized that: (i) antibiotic tolerance evolution could be constrained in the presence of a competitor but promoted in the presence of multiple antibiotics if cross-tolerance evolution is common and (ii) antibiotic exposure could change the community composition due to differences between the species' innate susceptibilities to the antibiotics or due to the evolution of tolerance-growth trade-offs.

We found that increases in antibiotic tolerance or the MIC were not generally enhanced by antibiotic combinations. Rather, increases in the tolerance or the MIC to a given antibiotic were driven by the presence of that antibiotic in the treatment combination, which occasionally led to cross-tolerance. Similarly, the presence of *S. maltophilia* did not affect the evolution of antibiotic tolerance or the MIC with either of the *P. aeruginosa* strains, likely due to the frequent extinctions during the early phase of the selection experiment. However, while both of the *P. aeruginosa* strains were able to dominate the "No Antibiotic" control and many of the antibiotic-containing coculture

treatments, the tobramycin-containing antibiotic treatments triggered *P. aeruginosa* extinctions in 15% of the coculture populations. Such extinctions were more common when *S. maltophilia* was cultured with PAO1 than with LESB58. Together, these results suggest that the effects of sub-MIC antibiotic concentrations could be magnified in polymicrobial communities due to competition, an asymmetry in innate antibiotic tolerances, and the differential evolution of antibiotic tolerances and their associated costs.

## RESULTS

**Effects of antibiotic treatments on *P. aeruginosa* antibiotic tolerance and relative cost of tolerance.** To test potential tolerance evolution, we measured the evolved *P. aeruginosa* populations' abilities to grow (as quantified by the optical density at 600 nm [$OD_{600}$]) in the presence of the treatment concentration of each of the antibiotics and compared this to the measured growth without the antibiotic (see Materials and Methods). Some *P. aeruginosa* monoculture replicates were removed from the analyses due to contamination with *S. maltophilia* (see Materials and Methods). Moreover, as *P. aeruginosa* went extinct in some of the tobramycin-containing treatments (24/143 selection lines), the evolution of tolerance was compared using only the surviving treatment replicates.

First, the antibiotic tolerances of both strains were not affected by previous exposure to *S. maltophilia* ($P > 0.05$) (Fig. 2; Tables S1 and S2). With regard to the control treatments, in the case of the clinical isolate LESB58, the "No Antibiotic" control treatment resulted in increased susceptibility to antibiotics, relative to the ancestor, whereas the antibiotic treatments maintained the ancestral-level tolerances of ciprofloxacin and tobramycin. In contrast, the "No Antibiotic" control treatment of the lab strain PAO1 maintained ancestral-level tolerance, whereas the antibiotic treatments further increased the tolerance of the evolved isolates (Fig. 2).

For both strains, there was a significant increase in ciprofloxacin tolerance when treated with the ciprofloxacin (CIP) mono-, CIP+colistin (CST), and CIP+tobramycin (TOB) treatments, compared to the "No Antibiotic" control treatment, as well as with the CIP+CST+TOB treatment in LESB58 (*post hoc* pairwise comparisons, PAO1: $t[48] = 11.78$ [CIP]; 13.76 [CIP+CST]; 11.42 [CIP+TOB], $P < 0.001$; LESB58: $t[66] = 4.73$ [CIP]; 4.82 [CIP+CST], $P < 0.001$; $t[66] = 4.42$, $P = 0.001$ [CIP+TOB]; $t[66] = 3.46$, $P = 0.026$ [CIP+CST+TOB]) (Fig. 2A and B). Similarly, the TOB and CIP+TOB treatments significantly increased tobramycin tolerance in PAO1 (*post hoc* pairwise comparisons, $t[48] = 7.71$ [TOB]; 6.20 [CIP+TOB], $P < 0.001$) (Fig. 2E), whereas all of the tobramycin containing treatments significantly increased the tolerance in LESB58 (*post hoc* pairwise comparisons, $t[66] = 6.39$ [TOB]; 6.53 [CIP+TOB]; 6.92 [CST+TOB]; 6.51 [CIP+CST+TOB], $P < 0.001$) (Fig. 2F). In contrast, no colistin-containing treatment resulted in increased colistin tolerance in PAO1, whereas only the CST+TOB and CIP+CST+TOB treatments significantly increased colistin tolerance in LESB58, compared to the "No Antibiotic" control treatment (pairwise *post hoc* comparisons, $t[66] = 3.30$, $P = 0.043$ [CST+TOB]; $t[66] = 3.39$, $P = 0.032$ [CIP+CST+TOB]) (Fig. 2C and D).

We also observed cross-tolerance between ciprofloxacin and tobramycin (i.e., the CIP mono-treatment provided tobramycin tolerance, and *vice versa*) (Fig. 2). In PAO1, TOB mono-treatment gave a significant increase in ciprofloxacin tolerance to a similar degree as that of the CIP mono-treatment, compared to the "No Antibiotic" control treatment (*post hoc* pairwise comparisons, $t[48] = 9.68$, $P < 0.001$) (Fig. 2A). Additionally, although not significant, the CIP mono-treatment increased the tobramycin tolerance (pairwise *post hoc* comparisons, $t[48] = 3.07$, $P = 0.093$) (Fig. 2E). A similar pattern emerged in LESB58, for which the CIP mono-treatment resulted in a significantly higher tobramycin tolerance than did the "No Antibiotic" control (*post hoc* pairwise comparisons, $t[66] = 3.73$, $P = 0.011$) (Fig. 2F). Further, although the TOB mono-treatment did not significantly increase ciprofloxacin tolerance, the CST+TOB treatment increased tolerance, compared to the "No Antibiotic" control (pairwise *post hoc* comparison, $t[66] = 3.94$, $P = 0.006$) (Fig. 2B). Additionally, the CIP+TOB combination treatment resulted in cross-tolerance to colistin in both strains (*post hoc* pairwise comparisons, PAO1: $t[48] = 5.04$, $P < 0.001$; LESB58: $t[66] = 4.21$, $P = 0.002$) (Fig. 2B and F). The CST mono-treatment did not provide any cross-tolerance toward the other

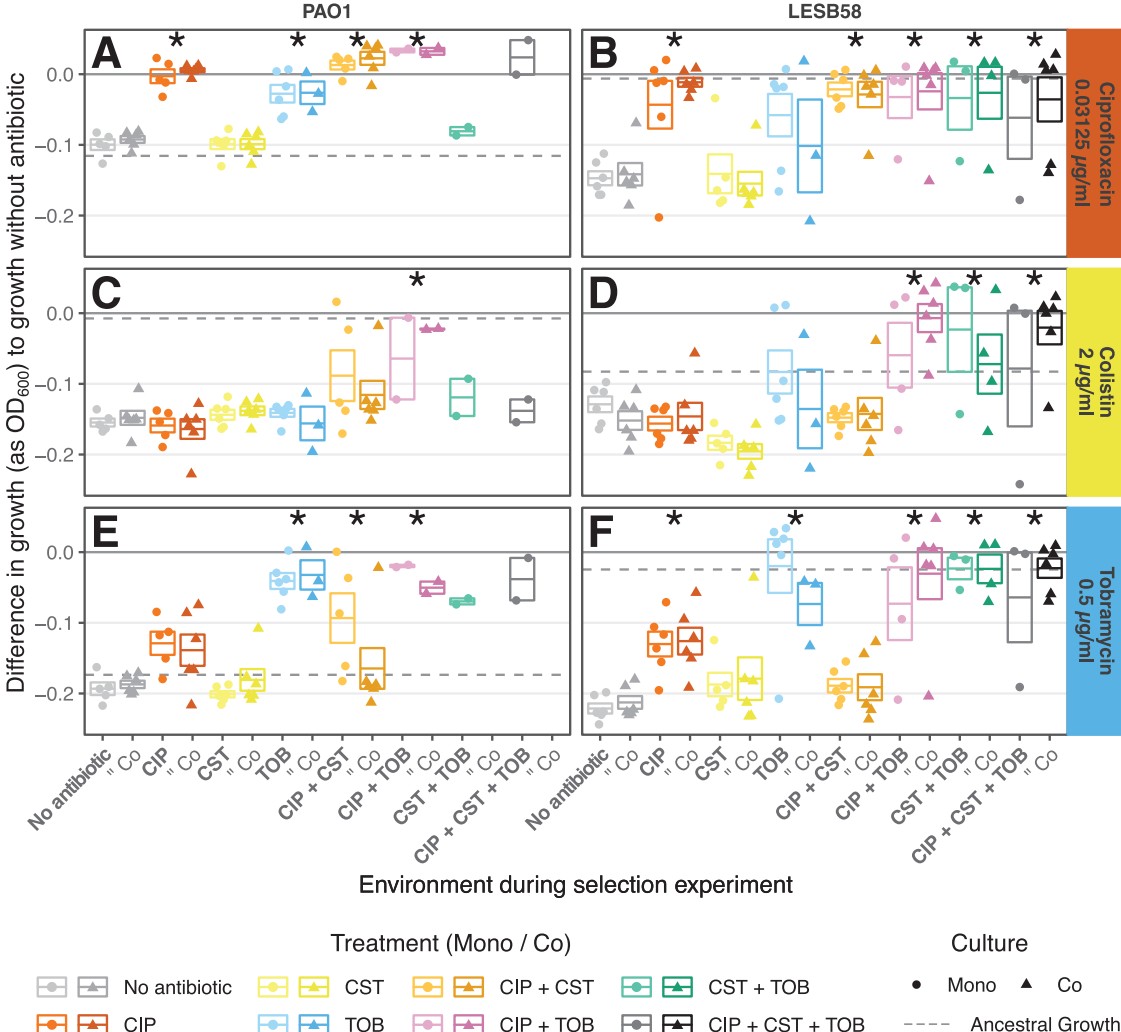

**FIG 2** Growth of each evolved *P. aeruginosa* replicate population in the treatment concentrations of antibiotic relative to their growth without antibiotic. Growth was measured in separate growth assays at the end of the selection experiment. The panel columns show the two *P. aeruginosa* strains, and the panel rows show the growth in the presence of different antibiotics. Each point represents the mean growth in antibiotic for three technical replicates of one replicate population, minus the growth without antibiotic of the same replicate population ($\Delta epOD_{600}^{Abx}$, as defined in Materials and Methods). The boxes show the means of all replicates (center line; $\Delta epOD_{600}^{Abx}$, as defined in Materials and Methods), the and upper and lower limits represent the ±SEM. The horizontal dashed line represents the $\Delta epOD_{600}^{Abx}$ of the ancestor. The solid dark gray line represents the growth equal to that without antibiotic (i.e., the relative change in $OD_{600}$ = 0). The shapes show monocultures (circle; "Mono") and cocultures (triangle; "Co"). The colors show antibiotic treatments, with lighter and darker shades representing the absence and presence of the *S. maltophilia* competitor, respectively. An asterisk indicates a statistically significant difference ($P < 0.05$) between the antibiotic treatment and the "No Antibiotic" control treatment via a *post hoc* pairwise comparison. A cross-tolerance between ciprofloxacin and tobramycin can be seen in panels A, B, E, and F, comparing the "No Antibiotic" treatment in columns 1 and 2 to the CIP and TOB treatments in columns 3, 4, 7, and 8.

antibiotics. These results suggest that while colistin tolerance evolution was rare, both pathogen strains readily evolved tolerance to ciprofloxacin and tobramycin, which was driven by prior exposure to these antibiotics and reciprocal cross-tolerance.

To test whether selection in different antibiotic treatments led to a cost of tolerance, we grew each of the surviving evolved replicates in media without antibiotics and compared their growth, relative to their respective ancestors (Fig. 3A and B). Across all treatments, the majority of both *P. aeruginosa* genotype replicates evolved to grow better in the growth media, relative to their ancestors (Fig. 3A and B). The increase in growth, relative to the ancestor, was greater in the lung-adapted LESB58 than in the lab-adapted PAO1. However, this increase clearly varied between the antibiotic treatments. In the case of both genotypes, the TOB mono-treatment constrained

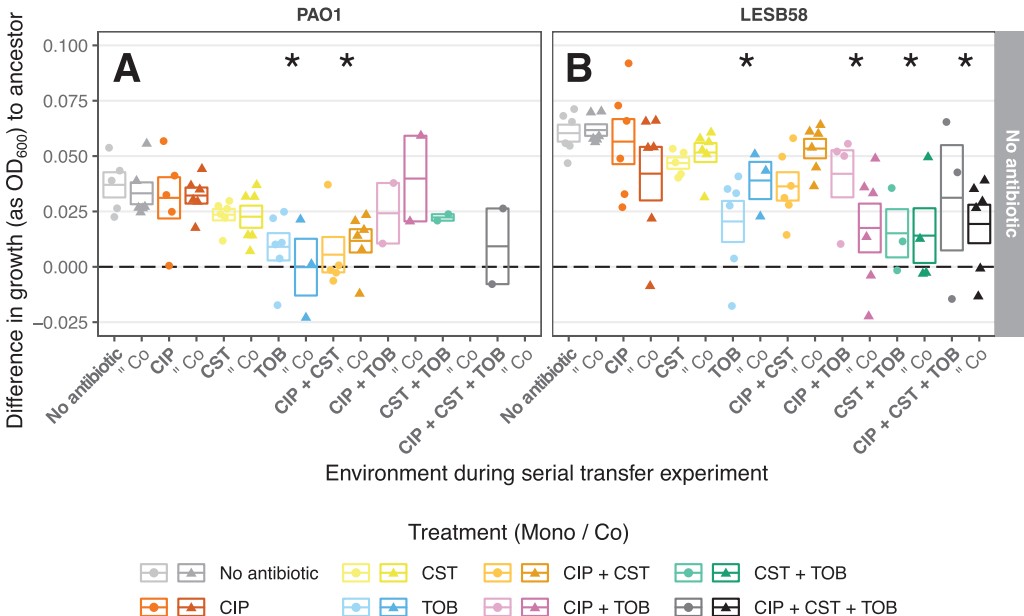

**FIG 3** Growth of each evolved *P. aeruginosa* replicate population without antibiotic relative to the respective ancestor. Growth was measured in separate growth assays at the end of the selection experiment. The panel columns show the two *P. aeruginosa* strains. Each point represents the mean growth without antibiotic for three technical replicates of one replicate population, minus growth of the ancestor under the same conditions ($\Delta epOD_{600}^E$, as defined in Materials and Methods). The boxes show the means of all replicates (center line; $\overline{\Delta epOD_{600}^E}$, as defined in Materials and Methods), and the upper and lower limits represent the ±SEM. The horizontal dashed line represents the growth equal to that of the ancestor (i.e., the relative change in $OD_{600} = 0$). The shapes show monocultures (circle; "Mono") and cocultures (triangle; "Co"). The colors show antibiotic treatments, with lighter and darker shades representing the absence and presence of the *S. maltophilia* competitor, respectively. An asterisk indicates a statistically significant difference ($P < 0.05$) between the antibiotic treatment and the "No Antibiotic" control treatment via a *post hoc* pairwise comparison. The cost of tobramycin tolerance can be seen in both panels by comparing the "No Antibiotic" treatment in columns 1 and 2 to the TOB treatments in columns 7 and 8.

adaptation, resulting in significantly reduced growth, compared to the "No Antibiotic" control treatment (*post hoc* pairwise comparisons, PAO1: $t[44] = 4.47$, $P = 0.001$; LESB58: $t[66] = 3.40$, $P = 0.03$). Moreover, the growth of evolved LESB58 populations treated with any tobramycin-containing antibiotic treatment were significantly below that of the "No Antibiotic" control treatment (*post hoc* pairwise comparisons, $t[66] = 3.58$, $P = 0.01$ [CIP+TOB]; 4.81, $P < 0.001$ [CST+TOB]; 3.89, $P = 0.005$ [CIP+CST+TOB]) (Fig. 3B). These results suggest that adapting to tolerate tobramycin reduced the growth and potential competitive ability of *P. aeruginosa* strains.

**Changes in the MIC of antibiotics with evolved *P. aeruginosa* populations.** We measured changes in the MIC of each antibiotic for evolved *P. aeruginosa* replicate populations as well as the MIC capable of inhibiting 50% of replicates ($MIC_{50}$) for each treatment to quantify whether exposure to low antibiotic concentrations led to an increased MIC. For both of the *P. aeruginosa* strains across all three antibiotics, there was no effect of previous exposure to *S. maltophilia* on the MICs. However, the MICs of the evolved populations changed considerably with both *P. aeruginosa* strains in response to all antibiotics (Table S3).

**(i) Changes in ciprofloxacin MIC.** Both evolved *P. aeruginosa* strains showed large increases in MIC to ciprofloxacin (Fig. 4A and B). The MIC of ciprofloxacin for the PAO1 replicates from the "No Antibiotic" control treatment remained mostly unchanged, relative to the ancestor, at 0.125 $\mu$g/mL, though a pair of individual replicates increased their MIC by 3-fold (Fig. 4A). In comparison, in LESB58, the baseline effect of the "No Antibiotic" control was a 3-fold decrease in MIC, compared to the ancestor, from 1 $\mu$g/mL to 0.125 $\mu$g/mL, the same MIC as that of the laboratory PAO1 strain (Fig. 4B). Pairwise chi-square tests showed that the CIP, CIP+CST, and CIP+TOB treatments all resulted in significantly greater MIC values, compared to the control treatment, among isolates of both strains (pairwise independence, PAO1: CIP: $\chi^2[1, N = 22] = 15.61$, $P = 0.002$; CIP+CST: $\chi^2[1, N = 22] = 10.77$, P = 0.010; CIP+TOB: $\chi^2[1, N = 15] = 8.34$, $P = 0.018$)

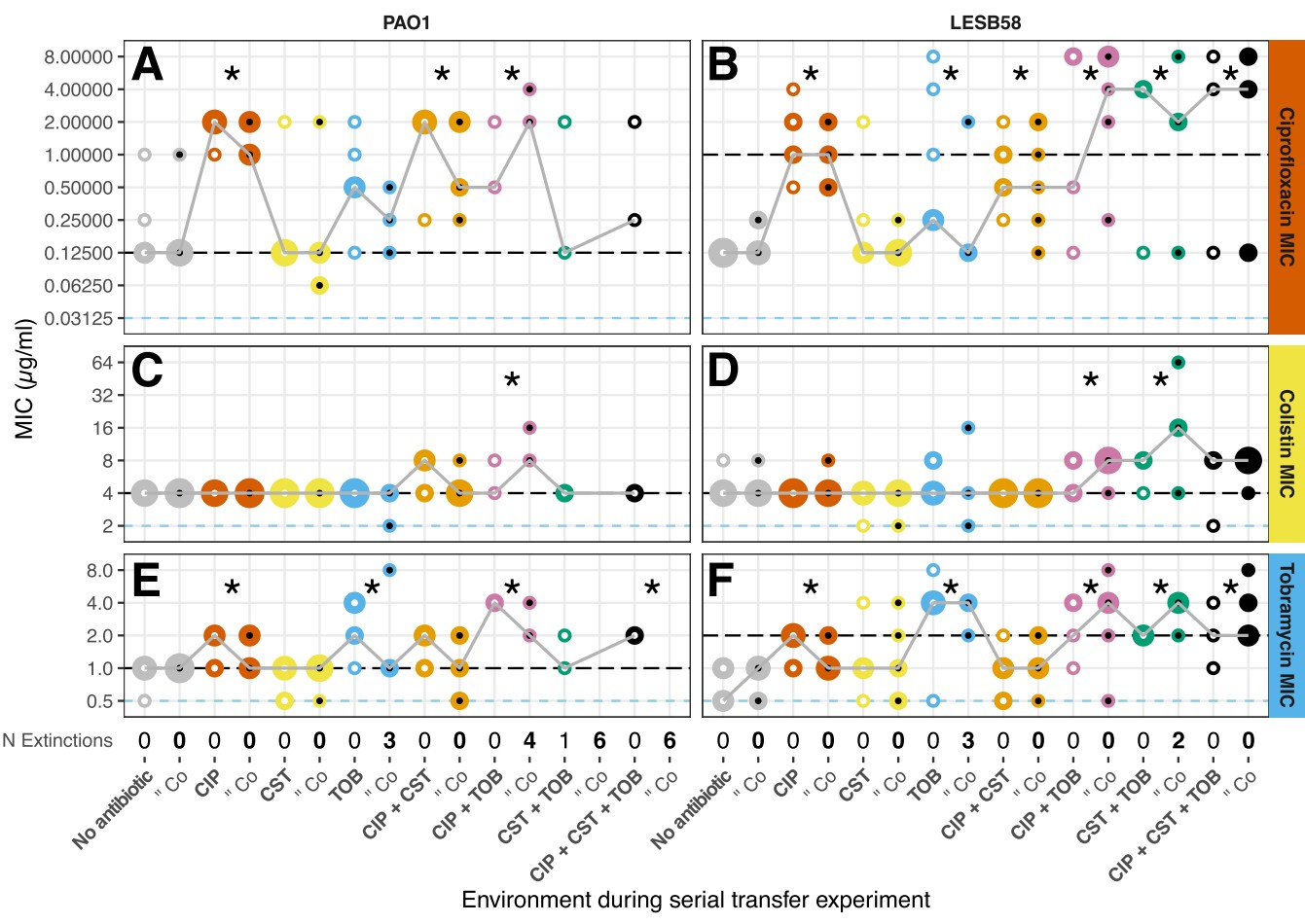

**FIG 4** The MIC of the three individual antibiotics for each evolved replicate population of the *P. aeruginosa* strains. The panel columns show the *P. aeruginosa* strain, and the panel rows show the MIC of each antibiotic. The dashed line represents the MIC of the respective ancestors. The gray line shows the $MIC_{50}$ of each treatment across replicate populations, as defined in Materials and Methods. The dashed blue line shows the treatment concentration. The size of each point represents the number of replicates with the specified MIC. The color of the points represents the treatment. The white center dot represents a monoculture, and black represents a coculture. The number of extinctions in each treatment and coculture is shown beneath the x axis. The MIC was measured in triplicate for each replicate. An asterisk represents a significant difference ($P < 0.05$) between the antibiotic treatment and the "No Antibiotic" control treatment via a *post hoc* pairwise independence test. A cross-tolerance between ciprofloxacin and tobramycin can be seen in panels A, B, E, and F, comparing the "No Antibiotic" treatment in columns 1 and 2 to the CIP and TOB treatments in columns 3, 4, 7, and 8.

(Fig. 4A); (pairwise independence, LESB58: CIP: $\chi^2[1, N = 24] = 19.03$, $P < 0.001$; CIP+CST: $\chi^2[1, N = 24] = 13.71$, $P = 0.0019$; CIP+TOB: $\chi^2[1, N = 22] = 13.31$, $P = 0.0019$) (Fig. 4B). Indeed, in LESB58, the triple antibiotic treatment also significantly increased the MIC values (pairwise independence: $\chi^2[1, N = 21] = 9.95$, $P = 0.0064$) (Fig. 4B), such that all ciprofloxacin-containing treatments increased the ciprofloxacin MIC. Moreover, many of the TOB mono-treated isolates from both strains had high MIC values, and, in LESB58, the MICs for both these and the CST+TOB treated isolates were significantly different from those of the "No Antibiotic" control treatment (pairwise independence, TOB: $\chi^2[1, N = 21] = 7.34$, $P = 0.019$; CST+TOB: $\chi^2[1, N = 19] = 10.03$, $P = 0.0064$), further suggesting that there is some cross-tolerance provided by tobramycin, as is also seen in the growth measurements (Fig. 3). The MIC values for the evolved LESB58 isolates reached higher levels than were observed in PAO1, with 18 LESB58 isolates reaching 4 or 8 μg/mL, compared with one PAO1 isolate. There was also greater variation in

the MIC values among the LESB58 isolates of a given treatment than among the PAO1 isolates of that treatment. Overall, both of the *P. aeruginosa* strains evolved an increase in the ciprofloxacin MIC, which was primarily driven by the previous exposure to ciprofloxacin.

**(ii) Changes in colistin MIC**. As opposed to ciprofloxacin, the MICs of colistin did not increase as a result of prior colistin exposure during the selection experiment (Fig. 4C and D). In the case of PAO1, the majority of treatments resulted in no change to the ancestral MIC of 4 $\mu$g/mL (Fig. 4C). Slightly more variation was observed among the LESB58 isolates that had been exposed to the CIP, CST, or TOB mono-treatments, though no changes greater than 1-fold for more than a single replicate were found. However, the MICs of isolates treated with the combinations CIP+TOB and CST+TOB were both significantly higher than that of the "No Antibiotic" control treatment (pairwise independence, CIP+TOB: $\chi^2[1, N = 22] = 6.13$, $P = 0.037$; CST+TOB: $\chi^2[1, N = 19] = 6.21$, $P = 0.037$) (Fig. 4C). Overall, only small changes in the colistin MIC were observed, and these were indirectly driven by other antibiotics.

**(iii) Changes in tobramycin MIC.** The MIC changes for tobramycin were similar between both *P. aeruginosa* strains (Fig. 4E and F). The "No Antibiotic" control-treated isolates of PAO1 maintained the ancestral MIC of 1 $\mu$g/mL (Fig. 4E), whereas the "No Antibiotic" control-treated LESB58 isolates decreased in MIC, relative to their ancestor (from 2 $\mu$g/mL down to 0.5 and 1 $\mu$g/mL) (Fig. 4F). For both strains, the TOB and CIP+TOB treatments resulted in a significant increase in MIC, compared with the "No Antibiotic" control treatment (pairwise independence, PAO1: TOB: $\chi^2[1, N = 20] = 8.54$, $P = 0.0035$; CIP+TOB: $\chi^2[1, N = 15] = 12.00$, $P < 0.001$) (Fig. 4E) (pairwise independence, LESB58: TOB: $\chi^2[1, N = 21] = 12.67$, $P = 0.0035$; CIP+TOB: $\chi^2[1, N = 22] = 11.35$, $P = 0.0053$) (Fig. 4F). For LESB58, this was the also the case with the CST+TOB and triple antibiotic treatments (pairwise independence, CST+TOB: $\chi^2[1, N = 19] = 13.77$, $P = 0.0035$; CIP+CST+TOB: $\chi^2[1, N = 21] = 13.22$, $P = 0.0035$). Both strains also had a significant increase in MIC as a result of the CIP mono-treatment, compared to the "No Antibiotic" control treatment (pairwise independence, PAO: $\chi^2[1, N = 22] = 7.98$, $P = 0.019$; LESB58: $\chi^2[1, N = 24] = 10.58$, $P = 0.0064$). However, this increase did not reach the same values as did the tobramycin containing treatments; the MIC of the CIP mono-treated LESB58 isolates was significantly lower than those of the TOB, CST+TOB, and CIP+CST+TOB treatments (pairwise independence, TOB: $\chi^2[1, N = 21] = 7$, $P = 0.021$; CST+TOB: $\chi^2[1, N = 19] = 8.05$, $P = 0.014$; CIP+CST+TOB: $\chi^2[1, N = 21] = 6.64$, $P = 0.023$), suggesting that the cross-tolerance provided by ciprofloxacin was weaker than that of tobramycin. Overall, both *P. aeruginosa* strains evolved increases in the tobramycin MIC, which was primarily driven by the previous exposure to tobramycin during the selection experiment.

**The effects of antibiotic treatments on bacterial densities and coculture compositions.** We measured a proxy of total population density (OD$_{600}$) of each bacterial population at the final time point of the selection experiment to determine the extent to which antibiotics inhibited bacterial growth (measurements were taken at the population level and did not differentiate species frequencies). For both *Pseudomonas* strains, there was a significant effect of antibiotic treatment on the total population density ($P < 0.001$) (Table S4), and with the exception of the CST mono-treatment, antibiotics generally decreased the total population density, relative to the "No Antibiotic" control treatment (Fig. 5A–D). However, *post hoc* pairwise comparisons showed that this effect was driven by the cocultures, as none of the monocultures differed significantly between any of the treatments with either strain (Fig. 5A and B). The combination of CST+TOB was particularly effective in the cocultures of both strains, reducing the population density significantly, compared with the "No Antibiotic" control (*post hoc* pairwise comparison, PAO1: $t[65] = 4.21$, $P = 0.005$; LESB58: $t[71] = 3.98$, $P = 0.010$) (Fig. 5C and D). There was no effect of growing in monoculture versus coculture for PAO1 ($P > 0.05$) (Table S4), and although there was a significant effect in LESB58 ($P = 0.022$) (Table S4), this was likely driven by the large difference in population density between the two CST-mono-treated cultures (*post hoc* pairwise comparison, $t[71] = 3.62$,

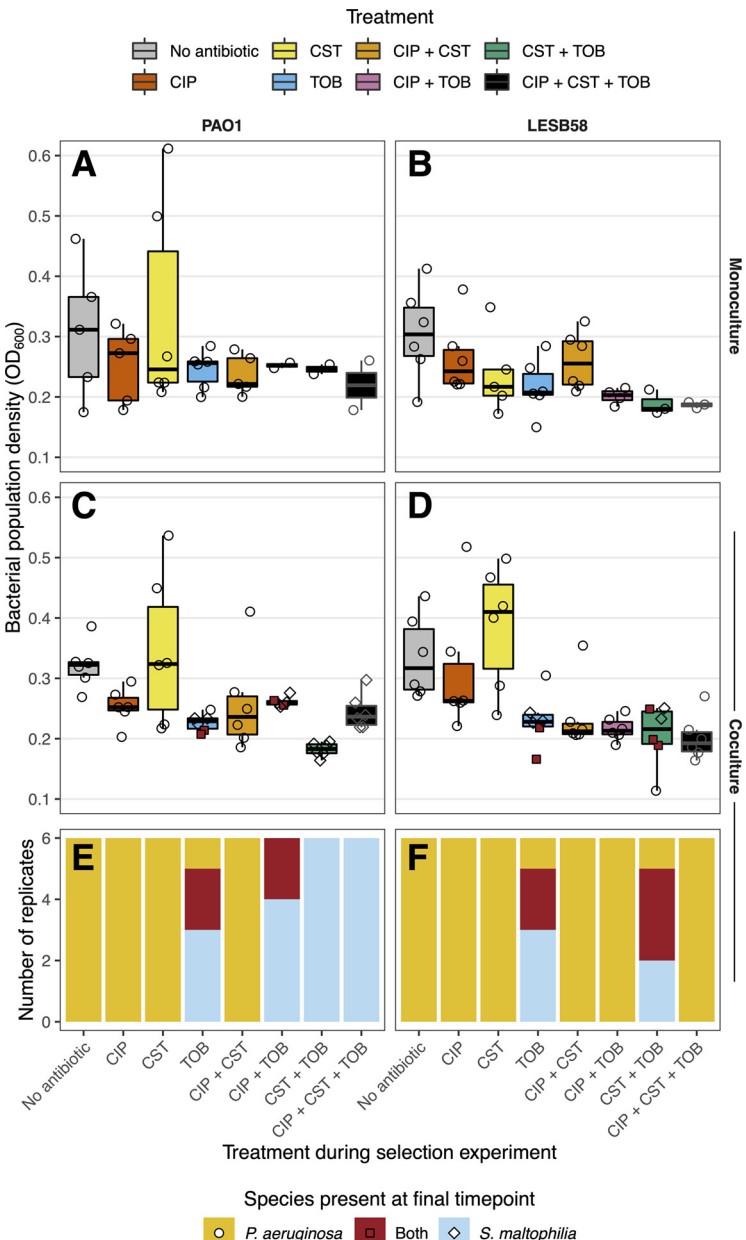

**FIG 5** Optical density of bacterial populations and composition of cocultures at the final time point of the selection experiment. (A–D) Boxplots of the optical density of the bacterial population (OD$_{600}$) of each replicate population from eight treatments (see the legend for the boxplot fill colors). The panel columns show the *P. aeruginosa* strain, and the panel rows show monocultures or cocultures. The points represent individual replicates (N = 6). The shapes show the species present at the final time point: *P. aeruginosa*, circles; *S. maltophilia*, diamonds; both, red squares. (E and F) The presence of surviving species in each coculture replicate (N = 6). The colors represent the surviving species as follows: *P. aeruginosa*, orange; *S. maltophilia*, blue; both, red.

$P = 0.034$) (Fig. 5B and D). Together, these results show that the antibiotics reduced the bacterial population densities, compared with the "No Antibiotic" control treatment, regardless of the presence of *S. maltophilia*, and that the combination of CST+TOB was highly effective at reducing the total bacterial population densities.

We also compared the composition of *P. aeruginosa* and *S. maltophilia* cocultures at the end of the experiment to examine the effects of antibiotics on the species coexistence. We found that *P. aeruginosa* survived in all monocultures across the different treatments (except for a single PAO1 replicate that went extinct under the CST+TOB treatment). This

suggests that the low concentrations of the antibiotic were not sufficient to kill *P. aeruginosa*, even when applied in combination. However, extinctions of *P. aeruginosa* were more common in the presence of *S. maltophilia* (Fig. 5E and F). In the absence of antibiotics, both of the *P. aeruginosa* genotypes were able to dominate the cocultures, driving *S. maltophilia* extinct in all replicates. The same was true in the CIP and CST mono-treatments. Moreover, in these treatments, *S. maltophilia* became undetectable at the early time points of the selection experiment (Fig. S2). In contrast, the TOB mono-treatment allowed for a prolonged coexistence between the two bacteria, and *S. maltophilia* was able to survive with both *P. aeruginosa* genotypes in two of the six TOB mono-treatment replicates and to fully outcompete *P. aeruginosa* in a further three replicates (Fig. 5E and F). The two *P. aeruginosa* genotypes differed in their capacity to coexist with *S. maltophilia* across the antibiotic combination treatments. The laboratory strain PAO1 was driven extinct more often than was the clinical strain LESB58; PAO1 was only able to survive in two CIP+TOB treated replicates and was otherwise driven extinct in the remaining 22 replicates across the other tobramycin-containing combination treatments (Fig. 5E). In contrast, the LESB58 strain dominated *S. maltophilia* in all of the combination treatments, except for the CST+TOB treatment, in which *P. aeruginosa* was driven extinct in two replicates (Fig. 5F). Together, these results suggest that while *P. aeruginosa* was able to outcompete *S. maltophilia* in most of the environments, this relationship was reversed in the presence of tobramycin, leading to either *P. aeruginosa* extinction or coexistence with *S. maltophilia*.

## DISCUSSION

While antibiotics are routinely used to treat *P. aeruginosa* infections within the polymicrobial communities in CF patient lungs, it is unclear how low antibiotic concentrations affect antibiotic resistance evolution in the presence of naturally occurring CF lung microbiota. Here, we studied this by exposing *P. aeruginosa* to sub-MICs of ciprofloxacin, colistin, tobramycin, and their combinations in the presence and absence of a commonly co-occurring CF species, *S. maltophilia*. We observed the tolerance evolution of *P. aeruginosa* to all antibiotics and a clear cross-tolerance between tobramycin and ciprofloxacin. Moreover, antibiotic exposure often led to an increase in the MIC, suggesting that sub-MIC selection can lead to high levels of antibiotic tolerance. While the presence of a competitor had no effect on the evolution of antibiotic tolerance, antibiotic exposure had a strong effect on the species community composition. Even though *P. aeruginosa* dominated most of the treatments, it either coexisted with or was driven into extinction by *S. maltophilia* in the presence of tobramycin, likely due to drastic innate differences in tobramycin resistance. Thus, even low doses of antibiotics could significantly change the evolution of antibiotic tolerance and the composition of multispecies communities.

We predicted that antibiotic tolerance evolution could be constrained by the presence of *S. maltophilia* as a competitor, for example, due to its negative effects on *P. aeruginosa* population densities and on the mutation supply rate of *P. aeruginosa* (27). However, we found that the presence of *S. maltophilia* did not alter the trajectory of antibiotic tolerance evolution. Instead, *P. aeruginosa* evolved increased tolerance to all antibiotics, regardless of the presence of the competitor, which was driven primarily by previous exposure to the same antibiotics during the selection experiment. The minimal effect of the competitor can, in some cases, be explained by the extinction of *S. maltophilia* early during the selection experiment, as was the case in the "No Antibiotic" control treatment and in the CIP and CST mono-treatments (Fig. S2). However, when *S. maltophilia* survived for longer, such as in the TOB mono-treatment or in the CST+TOB treatment for LESB58 (Fig. S2), there remained no significant effect of competition on the tolerance or the MIC evolution, compared with monoculture growth. This suggests that the presence of *S. maltophilia* indeed had no effect on *P. aeruginosa* tolerance evolution. However, due to the nature of our experimental setup, wherein evolutionary dynamics are inherently linked with ecological dynamics, it is difficult to isolate the effect that *S. maltophilia* may have had on the evolution of tolerance in *P. aeruginosa*. An alternative experimental setup, such as introducing fresh populations of *S. maltophilia* at each time point, could isolate the evolutionary effect of a

competitor. Alternatively, it is possible that the effect of *S. maltophilia* occurred at an earlier time point. Quinn et al. (48) found that the presence of *S. maltophilia* increased the rate at which imipenem resistance evolved in *P. aeruginosa* strain PAO1, compared to the *P. aeruginosa* monoculture. As such, although the final time point tolerances and MICs were similar between our monoculture and coculture *P. aeruginosa*, it is possible that they developed at differing rates.

We predicted that the presence of multiple antibiotics could select for increased levels of antibiotic tolerance, potentially due to cross-tolerance or via selection for generalized resistance mechanisms, such as efflux pumps (4, 5). In support of this, we found that increasing the numbers of antibiotics in the treatments resulted in higher level MICs, compared to mono-antibiotic treatments. We also found a reciprocal cross-tolerance between ciprofloxacin and tobramycin in both *P. aeruginosa* strains. We recognize that our measurements have some limitations, as our tolerance measurements were obtained using the optical density, which can be biased by changes in cellular morphology that are unrelated to tolerance evolution or by the presence of unviable cells. This bias is mitigated somewhat by calculating our proxy measurement, which compared growth with antibiotic to growth without (see Materials and Methods), as it compares the growth of the same replicate under different conditions, thereby accounting for any morphological changes that have arisen via adaptation to the growth medium. To fully compensate for this in the future, we could perform direct counts of viable cells, such as by counting colony-forming units (CFU).

While ciprofloxacin and tobramycin resistance can be mediated by the same mechanism in *P. aeruginosa*, namely, the upregulation of the MexXY-OprM efflux pump (49, 50), previous studies have suggested that *in vitro* selection for such mutations are rare (51, 52). As a result, other antibiotic-specific resistance mechanisms could have evolved, such as mutations in *fusA1* for tobramycin (52–54) and in *gyrAB* for ciprofloxacin (29, 55), even though these mutations are not known to provide cross-tolerance to the other antibiotic. Interestingly, in LESB58, the CST+TOB combination resulted in high levels of MICs for all three antibiotics, thereby providing a cross tolerance to ciprofloxacin. Colistin and tobramycin resistance can be mediated by outer membrane modifications via the activation of the PmrAB (56–58) and ParRS (59, 60) two-component systems. Gain-of-function mutations in either *pmrB* or *parS* can result in an increased tolerance to both antibiotics, and they have been observed both in *P. aeruginosa* treated with aminoglycosides *in vitro* (52, 61) and in the clinic (22, 58). However, whether the decrease in membrane permeability that these systems provide is sufficient to prevent the entry of ciprofloxacin is unclear. We also found that sub-MIC antibiotic selection often led to a clear increase in the MIC, which provided tolerance to much higher concentrations of antibiotics than the bacteria experienced during the selection experiment. With PAO1, this was especially clear with the ciprofloxacin and tobramycin MICs, whereas LESB58 showed an increase in MIC to all of the antibiotics, especially when exposed to antibiotic combinations. It has been shown previously that low levels of antibiotic selection can lead to high levels of resistance due to epistasis (20) and that antibiotic resistance can evolve *de novo*, even in the absence of antibiotic selection, due to adaptation to the growth medium (62). Further genetic analyses are of interest to ascertain the genetic mechanisms of antibiotic resistance at sub-MICs and to better understand the molecular basis of cross-tolerance.

We also predicted that antibiotic exposure could alter the community composition via potential differences in innate antibiotic sensitivity or tolerance-growth trade-offs. The baseline interaction between *P. aeruginosa* and *S. maltophilia* in the "No Antibiotic" control cocultures was antagonistic, whereby *P. aeruginosa* competitively excluded *S. maltophilia*. Indeed, it has previously been shown that *P. aeruginosa* can kill *S. maltophilia* via a contact-dependent mechanism during planktonic growth and that *P. aeruginosa* outcompetes *S. maltophilia* when grown in dual-species biofilms (63, 64). However, while the competitive exclusion of *S. maltophilia* was observed in the absence and presence of most of the antibiotic treatments, this pattern was reversed in the presence of tobramycin. In these treatments, we observed the coexistence of *P. aeruginosa* with the innately tobramycin-tolerant

*S. maltophilia* (65) or the extinction of *P. aeruginosa*. Though no follow-up data exist for the extinct *P. aeruginosa* replicates in the tobramycin-containing cocultures, it is possible that they were unable to evolve a tolerance to tobramycin, compared to the surviving replicates, and were thus competitively excluded from the cocultures. Moreover, the evolution of tolerance by *P. aeruginosa* did not restore competitive dominance, as the surviving coculture replicates that evolved tobramycin tolerance most frequently coexisted with *S. maltophilia*. The most probable explanation for this was the cost of tobramycin tolerance, which resulted in the reduced growth of the evolved *P. aeruginosa* isolates and likely led to less intense competition between the two species, although other mechanisms cannot be ruled out. We found that antibiotic treatments containing tobramycin were more effective at driving competitive exclusion in the naive PAO1 strain, suggesting that combination antibiotic treatments may be more effective at clearing *P. aeruginosa* than individual treatments. Of the antibiotic combinations, colistin and tobramycin together resulted in the extinction of both strains of *P. aeruginosa*, adding support for their efficacy in the clinic (15). This would be particularly important during early infection, as a failure to eradicate *P. aeruginosa* can select for mutations that aid in the establishment of long-term chronic infections (66). Together, our findings suggest that the efficacy of antibiotic combinations can be magnified in polymicrobial infections, which can lead to higher clearance of the target pathogen.

The competitive exclusion of *P. aeruginosa* differed between the two strains: the laboratory strain PAO1 was driven extinct in each tobramycin combination, except for two CIP+TOB replicates, whereas the clinical strain LESB58 was only affected in the CST+TOB treatment. A number of different factors may have contributed to this outcome. First, LESB58 had a relatively greater initial tolerance of ciprofloxacin and tobramycin (Fig. S1), which could have reduced the negative effects of the antibiotics in the CIP+TOB and triple antibiotic treatments, compared to PAO1. The initial differences in antibiotic susceptibility could reflect contrasting evolutionary histories between these two strains. PAO1 is highly lab-adapted due to repeated culturing in lab media, whereas LESB58, a transmissible strain isolated in 1988 from CF patients (67), is adapted to CF lungs in that it produces more biofilm than does PAO1 (68) and lacks motility (42). A further contributing factor is that LESB58 has a greater interbacterial competitive ability, meaning that it produces greater amounts of competitive factors, such as pyocyanin and proteases, and secretes these earlier in its growth phases (69, 70) than does PAO1, which is likely beneficial when in competition with other bacteria during polymicrobial CF infections (71, 72). Thus, LESB58 was likely preadapted to compete with other species, such as *S. maltophilia*, which could have contributed to the lower frequency of extinctions. Finally, in our experimental setup, the initial inoculant of LESB58 was 3-fold greater than that of PAO1, which may have increased the likelihood early in the selection experiment that LESB58 survived in the coculture treatments, whereas PAO1 did not. Further work would be required to disentangle each of these possibilities.

In summary, while the presence of the competitor did not affect the trajectory of antibiotic tolerance evolution in *P. aeruginosa*, we found that exposure to sublethal antibiotic concentrations led to more frequent extinctions of the target pathogen in the presence of an antibiotic-resistant competitor. Specifically, tobramycin played a key role in this eco-evolutionary process, in which the negative effect of this antibiotic persisted despite *P. aeruginosa* tolerance evolution, likely due to the associated growth costs. Somewhat worryingly, the *P. aeruginosa* populations often evolved increases in MIC to all antibiotics, leading to resistance against much higher concentrations of antibiotics than were experienced during the selection experiment. In conclusion, our results suggest that differences in antibiotic susceptibility can magnify competition in bacterial communities, leading to changes in community composition. The efficiency of the antibiotic treatment is then determined by both the surrounding community and the efficacy of delivery, choice of antibiotic, and antibiotic concentration, further complicating treatment design.

**TABLE 1** MICs for the three ancestral bacterial strains, along with the experimental concentrations used for each antibiotic

| Antibiotic | Treatment concentration ($\mu$g/mL) | MIC ($\mu$g/mL), (treatment concentration as a proportion of the MIC) | | |
|---|---|---|---|---|
| | | PAO1 | LESB58 | *S. maltophilia* |
| Ciprofloxacin | 0.03125 | 0.0625, (1/2) | 1, (1/32) | 0.125, (1/4) |
| Colistin | 2 | 4, (1/2) | 4, (1/2) | 8, (1/4) |
| Tobramycin | 0.5 | 1, (1/2) | 2, (1/4) | 8, (1/16) |

## MATERIALS AND METHODS

**Bacterial strains and culture conditions.** Two strains of *Pseudomonas aeruginosa* were used as the focal pathogen species: PAO1, a lab adapted reference strain (ATCC 15692), and LESB58, a transmissible CF lung isolated strain (42). *Stenotrophomonas maltophilia* type strain ATCC 13637 that had been isolated from the oropharyngeal tract of a cancer patient (73), was used as the coculture competitor. The base media used throughout was a 50:50 mix of nutrient broth without NaCl (Sigma; 5 g/L peptic digest of animal tissue, 3 g/L beef extract, pH 6.9) and PBS (8 g/L NaCl, 2 g/L KCl, 1.42 g/L $Na_2HPO_4$, 2.4 g/L $KH_2PO_4$), referred to here as "NB", that allowed for the stable coexistence of both the focal pathogen (*P. aeruginosa*) and the competitor (*S. maltophilia*) species over a single 72 h growth period. All cultures, unless otherwise stated, were grown at 37°C with shaking at 180 rpm.

**Selection experiment.** During the selection experiment, a focal bacterium (either *P. aeruginosa* strain PAO1 or LESB58) was grown in a culture either alone (monoculture) or with *S. maltophilia* (coculture) and was treated with subinhibitory concentrations of ciprofloxacin (CIP), colistin (CST), and tobramycin (TOB) antibiotics in all one-, two-and three-way combinations ("No Antibiotic," CIP, CST, TOB, CIP+CST, CIP+TOB, CST+TOB, CIP+CST+TOB). Each treatment was replicated 6 times for both focal pathogens in the absence and presence of *S. maltophilia*, resulting in a total of 192 selection lines. During the initial setup, overnight cultures from frozen stocks of PAO1, LESB58, and *S. maltophilia* were diluted down to the same optical density at 600 nm ($OD_{600}$; approximately 0.17 at 600 nm), corresponding to cell densities of $7.4 \times 10^6$, $2.2 \times 10^7$, and $4.6 \times 10^6$ CFU/mL, respectively. Monocultures consisted of 20 $\mu$L of the *P. aeruginosa* strain, whereas the cocultures mixed 10 $\mu$L of the *P. aeruginosa* strain with 10 $\mu$L of *S. maltophilia*, each in 200 $\mu$L of NB supplemented with 1 of the 8 antibiotic treatments for a total volume of 220 $\mu$L. The concentrations of the antibiotics (0.03125 $\mu$g/mL ciprofloxacin [Sigma-Aldrich], 2 $\mu$g/mL colistin [Acros Organics], and 0.5 $\mu$g/mL tobramycin [Acros Organics]) were chosen to be below the MIC of all three species (as determined below) (Table 1; Fig. S1), and were kept constant across each combination. Only one concentration of each antibiotic was used to model antibiotic therapy in the clinic, where antibiotics are applied without necessarily knowing the variation in the existing levels of resistance within the infecting populations. Selection lines were grown in 96-well plates. Then, setup plates were incubated for 72 h, after which each replicate was homogenized by mixing, and the $OD_{600}$ was measured (Tecan Infinite 200) as a proxy measure of the bacterial population densities. After measurement, each replicate was again mixed, and 20 $\mu$L of each culture were transferred to 200 $\mu$L of fresh medium with the same antibiotic treatment. These fresh plates were incubated for 72 h until the next serial transfer. The presence or absence of each species in each culture was determined following each transfer by growing subsamples of the 72-hour cultures on different selective agar; *Pseudomonas* selective agar (Oxoid; *Pseudomonas* agar base: 16 g/L gelatin peptone, 10 g/L casein hydrolysate, 10 g/L potassium sulfate, 1.4 g/L magnesium chloride, 11 g/L agar, 1% vol/vol glycerol; *Pseudomonas* CN selective supplement: 200 $\mu$g/mL centrimide, 15 $\mu$g/mL sodium nalidixate), and *S. maltophilia* selective agar: LB agar (10 g/L tryptone, 5 g/L yeast extract, 5 g/L NaCl, 15 g/L agar) supplemented with 64 $\mu$g/mL tobramycin incubated at 30°C, rather than 37°C, as *S. maltophilia* is innately resistant toward tobramycin at 30°C (65). Some monoculture replicates were contaminated with *S. maltophilia* (14 PAO1 selection lines, and 9 LESB58 selection lines), and these were excluded from the analyses. Whole population bacterial samples were picked from the agar plates for each replicate, grown overnight in NB, and cryopreserved in 20% glycerol to be frozen at −80°C. The selection experiment was carried out for 21 days, equating to 6 serial transfers. See Fig. 1 for a schematic of the treatment and culture combinations as well as the experimental procedures.

**Determination of MIC and antibiotic tolerance.** Both prior to and following the selection experiment, the MIC of each of the three antibiotics, ciprofloxacin, colistin, and tobramycin, was determined via broth microdilution for the three bacteria strains. Briefly, overnight cultures from frozen samples were diluted 1:10 in phosphate buffered saline (PBS) and were further diluted 1:10 into NB with antibiotic concentrations ranging from 32 $\mu$g/mL to 0.015625 $\mu$g/mL ($2^5$ to $2^{-6}$) and grown in static conditions in triplicate. The $OD_{600}$ was measured after 24 h (Tecan Sunrise). The MIC was defined as the lowest concentration of antibiotic at which there was no growth. We found no considerable difference between shaken and static conditions when determining the MIC in our preliminary experiments. For the evolved strains, the $MIC_{50}$ of a bacterial population was defined as the MIC required to inhibit half of the replicates of that population. We also grew all bacteria (evolved and ancestral) without antibiotic in NB for 24 h, using the same protocol as described above.

To assess the difference in tolerance of the antibiotics for each individual evolved replicate and ancestral strain, at the treatment concentrations used during the selection experiments, we defined a growth proxy, $\Delta epOD_{600}^{Abx}$, as

$$\Delta epOD_{600}^{Abx} = epOD_{600}^{Antibiotic} - epOD_{600}^{NoAntibiotic},$$

where $epOD_{600}^{Antibiotic}$ is the endpoint $OD_{600}$ from the growth of the replicate after 24 h in a given antibiotic. The definition is analogous for the growth of the same replicate without antibiotic (i.e., in NB). Such a measure is a proxy for the growth of the bacteria that accounts for the differences in medium adaptation when comparing antibiotic tolerance. Accordingly, we use this measure and its associated statistics in Fig. 2. The tolerance of the evolved bacteria from a selection regimen as a whole was calculated as

$$\overline{\Delta epOD_{600}^{Abx}} = \frac{1}{N(All)} \sum_{All} \Delta epOD_{600}^{Abx},$$

where we calculate the errors in our measure by computing the standard error of the mean (SEM). The summation is over all relevant strains evolved in the selection regimen of interest (e.g., monoculture PAO1 treated with CIP) and $N \leq 6$ replicates, dependent on extinctions.

We used a similar growth proxy, $\Delta epOD_{600}^{E}$, to assess the difference in medium adaptation for each individual replicate, compared to the respective ancestor, defined as

$$\Delta epOD_{600}^{E} = epOD_{600}^{Evolved} - epOD_{600}^{Ancestral},$$

where $epOD_{600}^{Evolved}$ is the endpoint $OD_{600}$ from the growth of the evolved strain after 24 h without antibiotic. The definition is analogous for the ancestral strain. We use this measure and its associated statistics in Fig. 3. We calculate the mean and SEM of $\overline{\Delta epOD_{600}^{E}}$ for a given selection regimen as with $\overline{\Delta epOD_{600}^{Abx}}$ above.

**Statistical analyses.** All of the data were analyzed in R version 4.1.0 (74). Data manipulation and graphing were performed using the *tidyverse* suite of packages (75), along with *egg* for figure assembly (76), and *ggbeeswarm* for point plotting (77). Regarding the tolerance data set, separate linear regression models were used for each *P. aeruginosa* species when analyzing each response variable (i.e., $\Delta epOD_{600}^{Abx}$ for the growth in each antibiotic, and $\Delta epOD_{600}^{E}$ for the growth without antibiotic). Here, the response variable was the difference in the growth in antibiotic, relative to the growth without antibiotic ($\Delta epOD_{600}^{Abx}$) for the antibiotic tolerance (Fig. 2), and the difference in growth relative to the ancestor ($\Delta epOD_{600}^{E}$) for the medium adaptation (Fig. 3). A two-way type II analysis of variance (ANOVA) was performed using the *car* package (78). *Post hoc* pairwise comparisons were computed from estimated marginal means, and *P* values were adjusted using the Šidák correction via the emmeans and contrast functions of the *emmeans* package (79). Pairwise comparisons were computed between treatments alone, after observing neither an effect of the competitor nor an interaction.

Regarding the MIC data set, individual Pearson's chi-square tests of independence were performed for each *P. aeruginosa* strain in each antibiotic. The MIC values were represented as ordered nominal variables, and the frequency of the observed MIC for each replicate in each treatment was tabulated. The chi-square tests were computed using the chisq_test function from the *coin* package (80). Pairwise tests of independence with the Benjamini-Hochberg false discovery rate corrections were performed between each treatment using the pairwiseOrdinalIndependence function from the *rcompanion* package (81).

With the population density data set, two linear regression models were fit, one to each *P. aeruginosa* strain. The response variable was the natural logarithm transformed $OD_{600}$ values, and the antibiotic treatment and the competitor were the predictor variables. A two-way type II ANOVA and *post hoc* pairwise comparisons were performed as with the tolerance data.

**Data availability.** The population density and species presence and absence data from the transfer experiment, the ancestral and evolved strain growth with and without antibiotic, and the MIC data have been deposited in Dryad (https://doi.org/10.5061/dryad.83bk3j9tn).

## SUPPLEMENTAL MATERIAL

Supplemental material is available online only.
**SUPPLEMENTAL FILE 1**, PDF file, 0.1 MB.

## ACKNOWLEDGMENTS

All authors conceived and designed the study. J.P.L. collected and analyzed the data. All authors wrote the manuscript.

This work was funded by the James Burgess Scholarship Ph.D. studentship at the University of York, awarded to J.P.L.

We declare that there are no conflicts of interest.

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
