## [Reviewer comments · Microbiology Spectrum]

Microbiology Spectrum

The effects of antibiotic combination treatments on *Pseudomonas aeruginosa* tolerance evolution and coexistence with *Stenotrophomonas maltophilia*

Jack Law, A. Jamie Wood, and Ville Friman

Corresponding Author(s): Ville Friman, University of York

Review Timeline:

Submission Date:	May 31, 2022
Editorial Decision:	July 14, 2022
Revision Received:	October 21, 2022
Accepted:	November 9, 2022

Editor: Olaya Rendueles Garcia

Reviewer(s): The reviewers have opted to remain anonymous.

Transaction Report:

DOI: <https://doi.org/10.1128/spectrum.01842-22>

July 14, 2022

Dr. Ville Friman
University of York
United Kingdom

Re: Spectrum01842-22 (The effects of antibiotic combination treatments on *Pseudomonas aeruginosa* tolerance evolution and coexistence with *Stenotrophomonas maltophilia*)

Dear Dr. Ville Friman:

Thank you for submitting your manuscript to Microbiology Spectrum.
First of all, allow me to excuse myself for the delay.
Two expert reviewers have read your manuscript and believe it is interesting.
But some changes have to be made prior to acceptance.

Link Not Available

Sincerely,

Olaya Rendueles Garcia

Journals Department
Reviewer comments:

Reviewer #1 (Comments for the Author):

Summary

In their paper-"The effects of antibiotic combination treatments on *Pseudomonas aeruginosa* tolerance evolution and coexistence with *Stenotrophomonas maltophilia*"-Law et al used an experimental evolution scheme to probe how different antibiotic regimes and interbacterial competition shape the evolution of *Pseudomonas aeruginosa* drug tolerance. The author's overall approach was straightforward: they passaged either 'laboratory' or 'clinical' strains of *P. aeruginosa* in the presence or absence of a competing strain of *Stenotrophomonas maltophilia* under various sub-MIC antibiotic regimes for a total of 6 serial passages spanning 21 days. After the final serial passage, the authors assessed multiple evolved lines/isolates for changes in

overall growth and antibiotic tolerance.

The strength of the author's study is that they are tackling the challenging problem of modeling the complex interactions between competing bacteria-in this case the cystic fibrosis-associated species *P. aeruginosa* and *S. maltophilia*-in the context of diverse antibiotic classes; specifically, ciprofloxacin (fluoroquinolone), colistin (polymyxin), and tobramycin (aminoglycoside). The authors ultimately conclude that the presence of *S. maltophilia* does not alter the potential of *P. aeruginosa* to evolve antibiotic tolerance. Although, in multiple cases and in the context of certain antibiotics, *S. maltophilia* drove *P. aeruginosa* to extinction. The clearest outcome from this study is that the evolution of tolerance to a specific antibiotic is largely induced by previous selection by the same antibiotic.

Although the topic and questions the authors address in their study are of general interest, the technical execution of experiments and presentation of data makes their findings difficult to interpret. Below I describe my main criticisms.

Major Comments

1. I think it is absolutely essential that the authors include a detailed schematic describing the cultivation and serial passaging steps used in their experimental evolution scheme. I found it incredibly difficult to understand from the text alone.
2. In general, I do not disagree with the author's approach for determining MICs for each antibiotic; however, they note on line 195 that MICs were determined using cultures grown statically for 24h. In contrast, during the experimental passaging scheme, it seems that cultures were grown with shaking (although it is unclear; for example, the details provided on lines 152 and 172 suggest shaking). Nonetheless, static growth will alter metabolism and growth rate and thus, could dramatically alter the MIC of antibiotics. Ideally, I would expect the conditions for determining MIC to closely match those used in the actual passaging experiment so that the level of selection pressure is consistent.
3. It is unclear why a single MIC was used for both *P. aeruginosa* strains. I appreciate the importance of comparing 'clinical' and 'lab-adapted' strains. But since PAO1 and LESB58 display such different intrinsic tolerances, using the same MIC for both would apply a fundamentally different selection pressure on each of them and thus, potentially induce different evolutionary trajectories. The patterns of evolution would be much easier to interpret if MICs were adjusted for each strain.
4. Related to comment 3, the authors clearly observed that each *P. aeruginosa* strain became adapted to growth in nutrient broth after experimental passaging (Figure 1G and H, gray boxplots). The co-selection of growth along with antibiotics and the presence of *S. maltophilia* makes it difficult to interpret the data and the authors do not consider their results in the context of adaptation to growth in nutrient broth. Ideally, to remove these confounding variables, the authors might want to consider pre-adapting their strains to nutrient broth prior to experimental passaging with antibiotics (for example, as was done here with *Acinetobacter baumannii* <https://elifesciences.org/articles/47612>).
5. To interpret the outcomes of the passaging experiments (especially for co-culture experiments), it would be helpful to see growth curve data and maximal growth rate measurements for each bacterium in each condition.
6. A major question proposed in the author's manuscript is whether the presence of *S. maltophilia* affects the evolvability of *P. aeruginosa*. However, no data is clearly presented that shows *S. maltophilia* was indeed present across all serial passages, or if it went extinct after the first round of passaging. These data should be included so that the results can be accurately interpreted. If *S. maltophilia* went extinct early on in the passaging scheme, then this would explain why the authors were not able to detect an interaction between *P. aeruginosa* and *S. maltophilia*.
7. Throughout the manuscript, but particularly in Figure 3, optical density is used as the only measure of growth. However, optical density cannot account for changes in cell size, which is known to be affected by antibiotic treatment (eg, via filamentation) and experimental passaging. To make any conclusions about 'Total bacterial population densities' (see line 826), the authors would need to include colony forming unit measurements that reflect the number of viable bacterial cells.

Reviewer #2 (Comments for the Author):

In their study "The effects of antibiotic combination treatments on *Pseudomonas aeruginosa* tolerance evolution and coexistence with *Stenotrophomonas maltophilia*" Law et al. evaluate the effect of Sub-MIC concentrations of antibiotics on the evolution of antibiotic resistance and tolerance in common cystic fibrosis pathogen *P. aeruginosa* grown in either mono or co-culture with competitor *S. maltophilia*.

The investigated research question is worthy of investigation and the selected methodological experimental evolution approach is albeit relatively simple in general appropriate to study it. The language of the manuscript is overall clear and understandable, however, throughout there are several very long sentences that could be separated in two to allow for a better logical flow of the arguments.

Overall, this manuscript presents relevant results in an understandable way. Still, a few major and minor points need to be addressed by the authors.

1. I question the calculation approach taken by the authors to assess if antibiotic tolerance evolved and what its cost is. The authors assess this by comparing OD as a growth proxy between the ancestral and the evolved populations. However, they simultaneously prove that even in the absence of a selective pressure the evolved strains have, by simply adapting to the growth medium, gained a significantly improved ability to grow in the medium. Consequently, the effects of adaptation to the growth medium and additionally gained tolerance are mixed in the authors approach. True gained tolerance can hence only be judged by comparing if the increase in relative growth at a certain treatment is bigger than that observed in the absence of antibiotics, rather than against 1 as the growth of the ancestral population. The current way of calculating it is artificially inflating the effect size.
2. In this figure 1 it would be beneficial to then mark those combinations with significant results using the * annotation
3. The inoculation size of LESB58 vs. PAO1 is increased by 3 fold. Does this have any effects on the subsequent measurements? This should be discussed.
4. I am wondering if the increased survival of LESB58 is indeed connected to a better adaptation to competition with other strains, as discussed by the authors. First the inoculation size as mentioned above is far increased, second its intrinsic tolerance to TOB is increased based on the data presented in SI1. These alternative interpretations need to be taken into account and judged.
5. Gaining molecular insights into the relevant mutations occurring in the population could be beneficial in the future. However, the story in this manuscript is in its current form coherent and self-contained and I would not see it as mandatory to be included. Just a suggestion for the future.

Staff Comments:

Preparing Revision Guidelines

Please return the manuscript within 60 days; if you cannot complete the modification within this time period, please contact me. If you do not wish to modify the manuscript and prefer to submit it to another journal, please notify me of your decision immediately so that the manuscript may be formally withdrawn from consideration by Microbiology Spectrum.

Summary

In their paper—“*The effects of antibiotic combination treatments on Pseudomonas aeruginosa tolerance evolution and coexistence with Stenotrophomonas maltophilia*”—Law et al used an experimental evolution scheme to probe how different antibiotic regimes and interbacterial competition shape the evolution of *Pseudomonas aeruginosa* drug tolerance. The author’s overall approach was straightforward: they passaged either ‘laboratory’ or ‘clinical’ strains of *P. aeruginosa* in the presence or absence of a competing strain of *Stenotrophomonas maltophilia* under various sub-MIC antibiotic regimes for a total of 6 serial passages spanning 21 days. After the final serial passage, the authors assessed multiple evolved lines/isolates for changes in overall growth and antibiotic tolerance.

The strength of the author’s study is that they are tackling the challenging problem of modeling the complex interactions between competing bacteria—in this case the cystic fibrosis-associated species *P. aeruginosa* and *S. maltophilia*—in the context of diverse antibiotic classes; specifically, ciprofloxacin (fluoroquinolone), colistin (polymyxin), and tobramycin (aminoglycoside). The authors ultimately conclude that the presence of *S. maltophilia* does not alter the potential of *P. aeruginosa* to evolve antibiotic tolerance. Although, in multiple cases and in the context of certain antibiotics, *S. maltophilia* drove *P. aeruginosa* to extinction. The clearest outcome from this study is that the evolution of tolerance to a specific antibiotic is largely induced by previous selection by the same antibiotic.

Although the topic and questions the authors address in their study are of general interest, the technical execution of experiments and presentation of data makes their findings difficult to interpret. Below I describe my main criticisms.

Major Comments

1. I think it is absolutely essential that the authors include a detailed schematic describing the cultivation and serial passaging steps used in their experimental evolution scheme. I found it incredibly difficult to understand from the text alone.

2. In general, I do not disagree with the author's approach for determining MICs for each antibiotic; however, they note on line 195 that MICs were determined using cultures grown statically for 24h. In contrast, during the experimental passaging scheme, it seems that cultures were grown with shaking (although it is unclear; for example, the details provided on lines 152 and 172 suggest shaking). Nonetheless, static growth will alter metabolism and growth rate and thus, could dramatically alter the MIC of antibiotics. Ideally, I would expect the conditions for determining MIC to closely match those used in the actual passaging experiment so that the level of selection pressure is consistent.
3. It is unclear why a single MIC was used for both *P. aeruginosa* strains. I appreciate the importance of comparing 'clinical' and 'lab-adapted' strains. But since PA01 and LESB58 display such different intrinsic tolerances, using the same MIC for both would apply a fundamentally different selection pressure on each of them and thus, potentially induce different evolutionary trajectories. The patterns of evolution would be much easier to interpret if MICs were adjusted for each strain.
4. Related to comment 3, the authors clearly observed that each *P. aeruginosa* strain became adapted to growth in nutrient broth after experimental passaging (Figure 1G and H, gray boxplots). The co-selection of growth along with antibiotics and the presence of *S. maltophilia* makes it difficult to interpret the data and the authors do not consider their results in the context of adaptation to growth in nutrient broth. Ideally, to remove these confounding variables, the authors might want to consider pre-adapting their strains to nutrient broth prior to experimental passaging with antibiotics (for example, as was done here with *Acinetobacter baumannii* <https://elifesciences.org/articles/47612>).
5. To interpret the outcomes of the passaging experiments (especially for co-culture experiments), it would be helpful to

see growth curve data and maximal growth rate measurements for each bacterium in each condition.

6. A major question proposed in the author's manuscript is whether the presence of *S. maltophilia* affects the evolvability of *P. aeruginosa*. However, no data is clearly presented that shows *S. maltophilia* was indeed present across all serial passages, or if it went extinct after the first round of passaging. These data should be included so that the results can be accurately interpreted. If *S. maltophilia* went extinct early on in the passaging scheme, then this would explain why the authors were not able to detect an interaction between *P. aeruginosa* and *S. maltophilia*.
7. Throughout the manuscript, but particularly in Figure 3, optical density is used as the only measure of growth. However, optical density cannot account for changes in cell size, which is known to be affected by antibiotic treatment (eg, via filamentation) and experimental passaging. To make any conclusions about 'Total bacterial population densities' (see line 826), the authors would need to include colony forming unit measurements that reflect the number of viable bacterial cells.

Response to Reviewers

We thank the Reviewers for their time and for their comments and suggestions. Please find our response below.

Reviewer #1

1. I think it is absolutely essential that the authors include a detailed schematic describing the cultivation and serial passaging steps used in their experimental evolution scheme. I found it incredibly difficult to understand from the text alone.

R1: We appreciate the difficulty in understanding the procedure for the experimental evolution experiment and have prepared a schematic to clarify the culture and antibiotic combinations used (new Figure 1, text reference on line 193–194 clean & tracked).

2. In general, I do not disagree with the author's approach for determining MICs for each antibiotic; however, they note on line 195 that MICs were determined using cultures grown statically for 24h. In contrast, during the experimental passaging scheme, it seems that cultures were grown with shaking (although it is unclear; for example, the details provided on lines 152 and 172 suggest shaking). Nonetheless, static growth will alter metabolism and growth rate and thus, could dramatically alter the MIC of antibiotics. Ideally, I would expect the conditions for determining MIC to closely match those used in the actual passaging experiment so that the level of selection pressure is consistent.

R2: As we had to quantify MIC for a high number of clones, we chose not to shake the plates during the incubation as we only had one plate shaker we could use. Despite the difference in shaken conditions to the selection experiment, we believe our conclusions to be valid as the same static condition was used to determine MIC for both the ancestral and evolved selection lines, and thus all comparisons between the two are with consistently determined values. Using preliminary data we found that shaking had only a very small effect on MIC values determined for PAO1 and LESB58; the two methods result in values that are either within 1-fold change of each other (ie 1 log₂ MIC) or the same (see Figures 1 & 2 and Table 1 below). We now justify this on lines 203–204 (242–243 tracked) in the Methods section.

Figure 1: Shaken MIC (Response letter only)

Figure 2: Static MIC (Figure S1 of manuscript)

Table 1: MIC values between growth methods (Response letter only)

	Shaken ($\mu\text{g/ml}$)	Nonshaken ($\mu\text{g/ml}$)
PAO1 CIP	0.0625	0.125
PAO1 CST	4	4
PAO1 TOB	2	1
LES CIP	1	1
LES CST	8	4
LES TOB	4	2

3. It is unclear why a single MIC was used for both *P. aeruginosa* strains. I appreciate the importance of comparing 'clinical' and 'lab-adapted' strains. But since PAO1 and LESB58 display such different intrinsic tolerances, using the same MIC for both would apply a fundamentally different selection pressure on each of them and thus, potentially induce different evolutionary trajectories. The patterns of evolution would be much easier to interpret if MICs were adjusted for each strain.

R3: Only one concentration of each antibiotic was used to model antibiotic therapy in the clinic where antibiotics are applied without necessarily knowing the variation in existing levels of resistance within the infecting populations. We now justify this on lines 172–175 (172–175 tracked) in the methods.

4. Related to comment 3, the authors clearly observed that each *P. aeruginosa* strain became adapted to growth in nutrient broth after experimental passaging (Figure 1G and H, gray boxplots). The co-selection of growth along with antibiotics and the presence of *S. maltophilia* makes it difficult to interpret the data and the authors do not consider their results in the context of adaptation to growth in nutrient broth. Ideally, to remove these confounding variables, the authors might want to consider pre-adapting their strains to nutrient broth prior to experimental passaging with antibiotics (for example, as was done here with *Acinetobacter baumannii* <https://elifesciences.org/articles/47612>).

R4: We note that Reviewer #2 made a similar comment (comment 9, below), and to address this issue we have recalculated tolerance as the difference between growth in antibiotic and without antibiotic, rather than growth of the ancestor in the same condition. The Methods (lines 208–221, 232–238 clean; 247–260, 284–291 tracked), Results (lines 278–312 clean; 333–367 tracked) and Figure 2 (previously 1) have been amended to match the new calculations. The results were largely similar and did not substantially change the conclusions. However, there were some small differences arising in terms of significant comparisons, which have been amended (e.g., the TOB mono-treatment no longer resulted

in significant ciprofloxacin tolerance for LESB58 [line 303 clean; 358 tracked]; there was also less support for the observation that colistin-containing treatments did not result in cross-tolerance [lines 308–309 clean; 363–364 tracked]).

We have retained the cost of tolerance section and made “growth relative to ancestor without antibiotic” a separate figure (now Figure 3). In line with this, the Methods have been updated (lines 222–228 clean; 261–271 tracked). When calculating the new statistics required for the updated tolerance section, the multiple comparisons adjustment was modified to the Šidák correction (line 240 clean; 294 tracked) and the p values on lines 313–328 (410–425 tracked) have been updated with the new correction (no change in significant differences).

5. To interpret the outcomes of the passaging experiments (especially for co-culture experiments), it would be helpful to see growth curve data and maximal growth rate measurements for each bacterium in each condition.

R5: Unfortunately, due to a high number of replicate populations, we were not able to collect growth curve data for all bacterial clones in each condition. While this data would potentially be interesting to show differences in growth dynamics, it should not change the main conclusions of this study.

6. A major question proposed in the author's manuscript is whether the presence of *S. maltophilia* affects the evolvability of *P. aeruginosa*. However, no data is clearly presented that shows *S. maltophilia* was indeed present across all serial passages, or if it went extinct after the first round of passaging. These data should be included so that the results can be accurately interpreted. If *S. maltophilia* went extinct early on in the passaging scheme, then this would explain why the authors were not able to detect an interaction between *P. aeruginosa* and *S. maltophilia*.

*R6: We have now included data on presence and absence of both species in all treatments throughout the selection experiment in Figure S2 (lines 431-432 clean; 572–573 tracked). These data indeed suggest that the minimal effect of *S. maltophilia* can be explained by extinction early in the experiment, as in the “No antibiotic” control, CIP and CST mono-treatments *S. maltophilia* was undetectable early on. In contrast, in the tobramycin-containing treatments where *S. maltophilia* survived for the majority of the experiment there was also little effect of *S. maltophilia* on tolerance evolution when compared with the monoculture. We discuss this and other possibilities on lines 470–487 (611–628 tracked) of the Discussion.*

7. Throughout the manuscript, but particularly in Figure 3, optical density is used as the only measure of growth. However, optical density cannot account for changes in cell size, which is known to be affected by antibiotic treatment (eg, via filamentation) and experimental passaging. To

make any conclusions about 'Total bacterial population densities' (see line 826), the authors would need to include colony forming unit measurements that reflect the number of viable bacterial cells.

R7: Due to a high number of replicate populations and evolved clones, we unfortunately had to rely on optical density measurements when estimating bacterial densities. We agree that OD values are difficult to interpret in the case of 'Total bacterial population densities' and that these measures could be biased by changes in the cell morphology, etc. We have amended the wording of the legend of Figure 5 (was 3) at line 890, (1136 tracked) and we raise the issue of potential bias on lines 493–501 (634–642 tracked) of the Discussion.

Reviewer #2

8. However, throughout there are several very long sentences that could be separated in two to allow for a better logical flow of the arguments.

R8: We appreciate this and have attempted to split and rearrange some of the longer sentences to improve flow.

9. I question the calculation approach taken by the authors to assess if antibiotic tolerance evolved and what its cost is. The authors assess this by comparing OD as a growth proxy between the ancestral and the evolved populations. However, they simultaneously prove that even in the absence of a selective pressure the evolved strains have, by simply adapting to the growth medium, gained a significantly improved ability to grow in the medium. Consequently, the effects of adaptation to the growth medium and additionally gained tolerance are mixed in the authors approach. True gained tolerance can hence only be judged by comparing if the increase in relative growth at a certain treatment is bigger than that observed in the absence of antibiotics, rather than against 1 as the growth of the ancestral population. The current way of calculating it is artificially inflating the effect size.

R9: Reviewer #1 made a similar suggestion (comment 4 above), and we have recalculated the results based on this suggestion. Please see response 4 for the full list of changes.

10. In this figure 1 it would be beneficial to then mark those combinations with significant results using the * annotation

*R10: We have added * to denote significant difference ($p < 0.05$) of an antibiotic treatment compared to the "No Antibiotic" control treatment by post-hoc pairwise comparison for figures 3, 4, & 5.*

11. The inoculation size of LESB58 vs. PAO1 is increased by 3 fold. Does this have any effects on the subsequent measurements? This should be discussed.

R11: We have checked the optical density measurements from the first timepoint of the selection experiment (shown below) and note that, at least in terms of optical density (which as discussed by Reviewer #1 is not perfect), there was no consistent significant difference between the two P. aeruginosa strain. (Populations that differed significantly between strains, by pairwise comparison: Mono CIP, Mono & Co CST, Co CIP+TOB, Co CST+TOB.) It is thus likely that over the course of the experiment these initial differences had little effect on tolerance/MIC evolution. However, we agree that this difference in inoculant size may play a role in the increased survival of LESB58, and have reframed the discussion of the differences between the two strains in terms of a number of contributing possibilities (lines 557–573 clean; 712–728 tracked).

Figure 3: Timepoint 1 densities (Response letter only)

12. I am wondering if the increased survival of LESB58 is indeed connected to a better adaptation to competition with other strains, as discussed by the authors. First the inoculation size as mentioned above is far increased, second its intrinsic tolerance to TOB is increased based on the data presented in SI1. These alternative interpretations need to be taken into account and judged.

R12: Please see response 11.

13. Gaining molecular insights into the relevant mutations occurring in the population could be beneficial in the future. However, the story in this manuscript is in its current form coherent and self-contained and I would not see it as mandatory to be included. Just a suggestion for the future.

R13: We agree and have taken this approach in follow up studies, and we have amended the wording slightly at line 523 (678 tracked) in line with this suggestion.

November 9, 2022

Dr. Ville Friman
University of York
United Kingdom

Re: Spectrum01842-22R1 (The effects of antibiotic combination treatments on *Pseudomonas aeruginosa* tolerance evolution and coexistence with *Stenotrophomonas maltophilia*)

Dear Dr. Ville Friman:

Thank you for your patience. I am glad to communicate that your manuscript has been accepted, and I am forwarding it to the ASM Journals Department for publication. You will be notified when your proofs are ready to be viewed.

Sincerely,

Olaya Rendueles Garcia
Editor, Microbiology Spectrum
